# Style Randomization Improves the Robustness of Breast Density Estimation in MR Images

**Goksenin Yuksel** [1]                                     Goksenin.yuksel@student.uva.nl
**Koen Eppenhof** [2]                                       koen.eppenhof@screenpointmed.com
**Marcel Worring** [3]                                                     m.worring@uva.nl
**Jaap Kroes** [4]                                          jaap.kroes@screenpointmed.com
[1,3] *UvA Amsterdam, Netherlands*
[2,4] *Screenpoint Nijmegen, Netherlands*

**Editors:** Accepted for publication at MIDL 2024

## Abstract

Breast density, a crucial risk factor for future breast cancer development, is defined by the ratio of fat to fibro-glandular tissue (FGT) in the breast. Accurate breast and FGT segmentation is essential for robust density estimation. Previous research on FGT segmentation in MRI has highlighted the significance of training on both images with and without fat suppression to enhance network's robustness. In this study, we propose a novel data augmentation technique to further exploit the multi-modal training setup motivated by the research in style randomization. We demonstrate that the network trained with the proposed augmentation is resilient to variations in fat content, showcasing improved robustness compared to solely training with multi-modal data. Our method effectively improves FGT segmentation, thereby enhancing the overall reliability of breast density estimation.

**Keywords:** Breast Density Estimation, MRI, Style Randomization, Deep Learning, FGT Segmentation, Robustness, Representation Learning, Dixon Images

## 1. Introduction

Breast density is defined as the amount of fibro-glandular tissue (FGT) in the breast, and it is a well-established risk factor for the future development of breast cancer (Boyd et al., 2007). While automated software solutions for breast density estimation are readily available for mammography, limited efforts have been directed towards creating an automated robust breast density calculation tool tailored for breast MRI (Chalfant and Hoyt, 2022). Yet, MRI allows for more accurate quantification of breast density compared to digital mammography (Gubern-Mérida et al., 2014).

There are multiple ways of performing breast density estimation in MRI (van der Velden et al., 2020). Recent literature in this area mostly focused on breast density estimation via segmentation (Doran et al., 2017; van der Velden et al., 2020). Volumetric breast density estimation is then calculated by comparing the number of voxels classified as FGT to the overall volume of the breast.

Automatic breast and FGT segmentation in breast MRI has been addressed by several papers (Dalmış et al., 2017; Huo et al., 2021; Zhang et al., 2019). Recent literature suggests that deep learning methods have consistently shown great results in automating various

medical imaging tasks, consistently outperforming traditional methods including breast-FGT segmentation.

Dalmış et al. (2017) were the first to propose a deep learning-based approach to breast density estimation, exceeding the performance of previously developed traditional machine learning-based approaches. Later, Zhang et al. (2019) extended the previous research on 2D U-Net for FGT segmentation using a multi-vendor dataset. Later, Huo et al. (2021) used a 3D U-Net, exceeding the performance of all known 2D U-Net architectures on FGT segmentation.

The papers mentioned above did not carry out testing on both fat-suppressed (FS) and without fat-suppressed (WOFS) acquisitions. To address this issue, Samperna et al. (2022) trained a 3D U-Net architecture on both FS and WOFS acquisitions. They utilized Dixon images in the training set to include both acquisitions of the same patient. The study demonstrates that incorporating both WOFS and FS acquisitions in the training set enables the achievement of similar results, while training exclusively on one modality leads to sub-optimal performance when tested on the other.

Sub-optimal results may be attributed to the style difference in the WOFS and FS images since Convolutional Neural Networks (CNNs) exhibit bias towards styles (Baker et al., 2018). This style bias does not arise from the inductive properties of the CNNs, but rather from the contents of the training data (Geirhos et al., 2018). The observed performance improvement in Samperna et al. (2022) may be attributed to the elimination of style bias by including different styles of the same patient.

The style bias can be further reduced by appearance-modifying data augmentation (Hermann et al., 2020). Xu et al. (2021) proposes to modify the amplitude information of images where mostly low-level signals are present, and keep the phase information intact where high-level semantics are present. This augmentation leads to a model that can better extract the semantic concepts, and is more robust to domain shift. Moreover Jackson et al. (2019) introduces style randomization as data augmentation. This technique incorporates various styles of the same image during training. This approach aids in developing a more robust model and shows enhanced performance across multiple domains without requiring specific data from each.

In this paper, we apply style randomization with Dixon images to enhance robustness of breast and FGT segmentation, and thus breast density estimation. We argue that this randomization technique introduces more style variability while keeping the high level semantics present. This method forces the network to be more robust by eliminating the style bias even further, and teaches the network to focus more on the shape rather than texture.

We find that style randomization helps the network to generalize better, and alleviates the style bias even further. Another important finding is that style randomization improves FGT dice score on both training and test sets. These findings imply that utilizing style randomization during the training not only boosts the robustness of the network but also helps adapt the network to various potential target domains.

## 2. Methods

**Dixon Images**

The Dixon method provides fat-water separation in MR images. The key idea behind Dixon imaging is to acquire multiple images at different echo times, exploiting the chemical shift difference between fat and water protons. Fat and water have slightly different resonant frequencies due to their different chemical compositions, and by acquiring images at specific echo times, it's possible to create separate images highlighting fat and water components (Dixon, 1984).

## 2.1. Style Randomization

By style randomization, we extend on the methodology from Samperna et al. (2022). Given that Dixon images contain water (W) and fat (F), we randomize the style of the image by introducing mix-up parameter $\alpha$. We call this **Mixed Modality**.

$$I = \begin{cases} W & \textbf{FS} \\ W + \alpha F & 0 < \alpha < 1 \\ W + F & \textbf{WOFS} \end{cases}$$

To randomize the style of an image we follow two strategies. First, we use alpha levels from [0, 0.2, 0.5, 0.8, 1]. Second, we use alpha levels from 0 to 1, with uniform steps of 0.1. From now on, for the sake of readability, we will refer to these as **Augmented** and **More Augmented** strategies. We use both strategies to quantify the effect of different style randomization levels. Style randomization is applied intra-patient to achieve linear mixture of WOFS and FS modalities. Figure 1 visualizes an MR acquisition augmented with a More Augmented strategy. The below visualization demonstrates that adding fat to the water image does not alter the global shape of the FGT, and the breast remains unchanged while mixing the WOFS and FS domains linearly. We can achieve the same training setup as Samperna et al. (2022) when we utilize style randomization with [0,1] fat levels

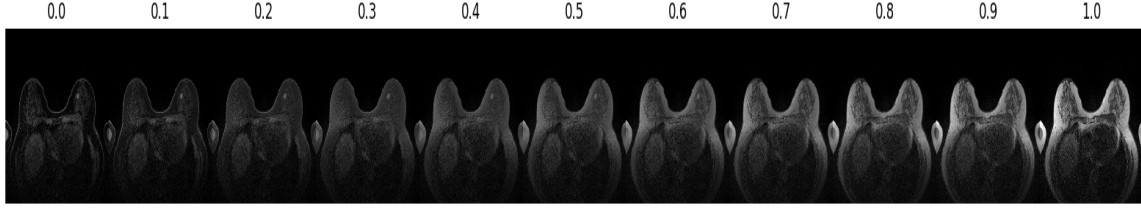

Figure 1: Gradient of fat augmentation, each title depicts the weight of fat in the image. When the image has 0 or 1 fat weight it results FS and WOFS images.

Additionally, we employ a test time augmentation for evaluation purposes. We randomly chose two fat fractions and augment one image. We refer to this strategy as **Randomly Weighted Fat**.

## 2.2. Model

We employ the nn-UNet framework (Isensee et al., 2021). We used patch-based full-resolution 3D U-Net for all our experiments. We treat each image as a separate sample while using the same ground truth segmentation for the same patients. The network's input was a single patch of size $192 \times 192 \times 64$, with a batch size of 2. We trained all networks for 1000 epochs with 250 iterations per epoch. The default augmentation strategy was applied with instance-based z-score normalization. No post-processing was applied.

## 2.3. Data

To implement the proposed method, we employed an in-house training set with Dixon images. They are acquired between the periods between 2015-2017. Breast MRI acquisitions were obtained using a 3T breast MRI scanner. All the acquisitions included in this study were pre-contrast T1-weighted acquisitions. We divided the dataset into training (n = 51 patients) and testing (n = 9 patients, 18 acquisitions (9 WOFS, 9 FS)). Additionally, we employed 5-fold cross-validation. Ground truth labels were artificially generated using a pre-trained network.

For independent test sets, we used an evaluation set published by Müller-Franzes et al. (2023). This is a subset of DUKE, which was collected between 2000 and 2014 at the Duke Hospital, USA, which is publicly available Saha et al. (2021). This dataset contains acquisitions from 40 patients. The acquisitions were obtained by a 1.5 Tesla, or 3.0 Tesla scanner from General Electric or Siemens. The MRI protocol consisted of a T1-weighted fat-suppressed sequence(one pre-contrast, and four post-contrast scans) and a non-fat-suppressed T1-weighted sequence. We used a pre-contrast T1-weighted fat-suppressed image to evaluate our algorithm. We refer to this test set as Müller-Franzes et al. (2023). Additionally, we utilized published breast segmentation and FGT segmentations by DUKE. This set contains 100 patients and has the same characteristics as the set described above. After processing the masks, we decided to exclude the ones that we could not parse in SITK. We also found that some images has more than 3 dimensions. Due to lack of metadata to guide us on how to use them, we decided to drop these as well. This left us with 88 patients. Later, we only used segmentation labels for breast and FGT. We refer to this test set as Saha et al. (2021).

For the BI-RADS density estimation task, we employed radiologists estimations from DUKE. This data contains BIRADS density label estimations from three radiologists for 50 instances. In all cases two out of three radiologists agreed on the density grade, with the third radiologist estimate differing at most one grade from the other. Therefore we utilized majority voting to consolidate the estimations into a single label.

## 3. Results

Table 1 demonstrates that mixed modality training does not show significant performance drift when tested on WOFS or FS modality. We visualize the lowest performance case from the test set. The visualization of the test sample is included in Table 2. This patient has a breast implant. In the WOFS image, we classify the implant as part of the FGT tissue, therefore breast density prediction is higher than the ground truth.

| Test | Augmentation Strategy | Breast DSC [95% CI] | FGT DSC [95% CI] | Pearson's r |
|------|----------------------|---------------------|------------------|-------------|
| WOFS | Augmented | 0.97 [0.96,0.98] | 0.93 [0.88,0.97] | 0.98 [0.95, 1.00] |
| WOFS | More Augmented | **0.99** [0.99,0.99] | **0.97** [0.96,0.98] | 1.00 [1.00, 1.00] |
| WOFS | Mixed Modality | 0.96 [0.94,0.98] | 0.91 [0.84, 0.95] | 0.92 [0.86, 1.00] |
| FS | Augmented | 0.95 [0.92,0.97] | 0.89 [0.86,0.92] | 0.99 [0.96, 1.00] |
| FS | More Augmented | **0.98** [0.97,0.98] | **0.90** [0.88,0.93] | 1.00 [1.00, 1.00] |
| FS | Mixed Modality | 0.95 [0.92, 0.97] | 0.87 [0.84, 0.91] | 0.99 [0.97 − 1.00] |
| Randomly Weighted Fat | Augmented | 0.97 [0.96,0.98] | 0.90 [0.86,0.93] | 0.97 [0.93,1.00] |
| Randomly Weighted Fat | More Augmented | **0.98** [0.98,0.99] | **0.94** [0.92,0.96] | **1.00** [1.00,1.00] |
| Randomly Weighted Fat | Mixed Modality | 0.94 [0.92,0.97] | 0.80 [0.71,0.87] | 0.96 [0.92, 1.00] |
| Saha et al. (2021) | Augmented | 0.87 [0.86,0.88] | **0.72** [0.70,0.75] | **0.98** [0.97,0.99] |
| Saha et al. (2021) | More Augmented | 0.87 [0.86,0.88] | **0.72** [0.69,0.74] | **0.98** [0.96,0.99] |
| Saha et al. (2021) | Mixed Modality | 0.87 [0.85,0.88] | 0.71 [0.68,0.74] | 0.97 [0.94,0.98] |
| Müller-Franzes et al. (2023) | Augmented | **0.84** [0.83,0.86] | **0.69** [0.64,0.73] | **0.92** [0.86-0.96] |
| Müller-Franzes et al. (2023) | More Augmented | **0.84** [0.83,0.85] | **0.69** [0.64,0.73] | **0.92** [0.85-0.96] |
| Müller-Franzes et al. (2023) | Mixed Modality | 0.83 [0.81,0.84] | 0.63 [0.58-0.68] | 0.84 [0.75-0.93] |

[**]$p < 0.05$, [*]$p < 0.1$

Table 1: Augmentation results. We use sign test to calculate P values. We compared only the results of independent test sets.

Moreover, Table 1 shows that style randomization improves on mixed-modality training in every dataset. Domain shift does not affect the performance of the networks trained with augmentations. We see that the test performance of mixed modality in Randomly Weighted Fat degraded heavily on FGT segmentation, while augmented networks are not affected by it. Moreover, in independent test sets, more augmented network significantly outperforms mixed modality training. More augmented network demonstrates a significant performance improvement over mixed modality training. Notably, the more augmented network not only suppresses the results of mixed modality and augmentation but also eliminates outliers, such as the case with a breast implant in the test set.

Figure 2 illustrates that more augmentation enhances the dice scores of individual cases by up to 0.4 compared to the baseline. Additionally, it reveals that augmentation may result in a deterioration in dice scores for certain cases. Nevertheless, the overall enhancement outweighs the instances of decline. Table 2 shows that failing segmentations from mixed-modality training are over segmented. Whereas a network trained on style randomized images correctly segments the FGT tissue, yielding a better FGT segmentation performance.

### 3.1. Numerical Density Estimation

Pearson's r from Table 1 shows that predicted density estimations from style randomized networks correlate significantly more with the ground truth compared to mixed modality training.

Figure 3 visualizes the Bland-Altman plots for numerical agreement between predicted density, and ground truth density. We observe that all models predict the breast density well. The mean disagreement falls between 0.02% to 0.04%. Furthermore, density predictions from networks trained with the proposed augmentation are more precise. The confidence intervals indicate that proposed augmentation has less variance in predicted density error, and is closer to 0. The outliers in the Bland-Altman plots are alleviated with the

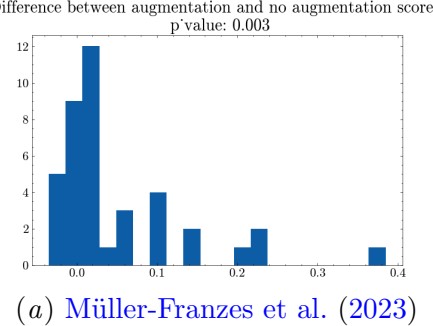

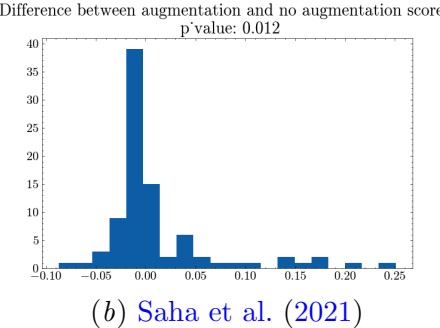

(a) Müller-Franzes et al. (2023)       (b) Saha et al. (2021)

Figure 2: Instance based dice score difference between mixed modality, and more augmented.

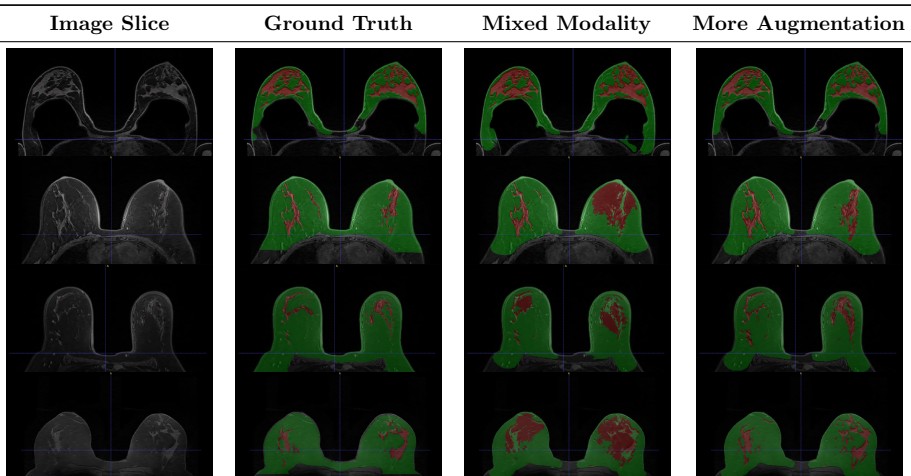

Table 2: Test acquisitions where we achieve the biggest performance improvements in FGT Dice score with style randomization. Red indicates FGT, and green indicates breast voxels. First row shows the acquisition with the breast implant. Following two rows are taken from Saha et al. (2021), and last row Müller-Franzes et al. (2023)

style randomization. Figure 3 depicts a positive correlation between the density of breast and error in predictions for (Saha et al., 2021).

## 3.2. BI-RADS Density Estimation

We quantify the agreement between radiologists and the density predictions from the network using the BI-RADS labeled data. Figure 4 depicts a box plot of BIRADS density labels to the numerical density estimation. All networks show a clear trend of increased density when from left to right agreeing with the radiologists.

Whisker in Figure 4 depicts that the network trained on mixed modality over-estimates the density. This phenomenon is especially visible for BIRADS label B. On the other hand,

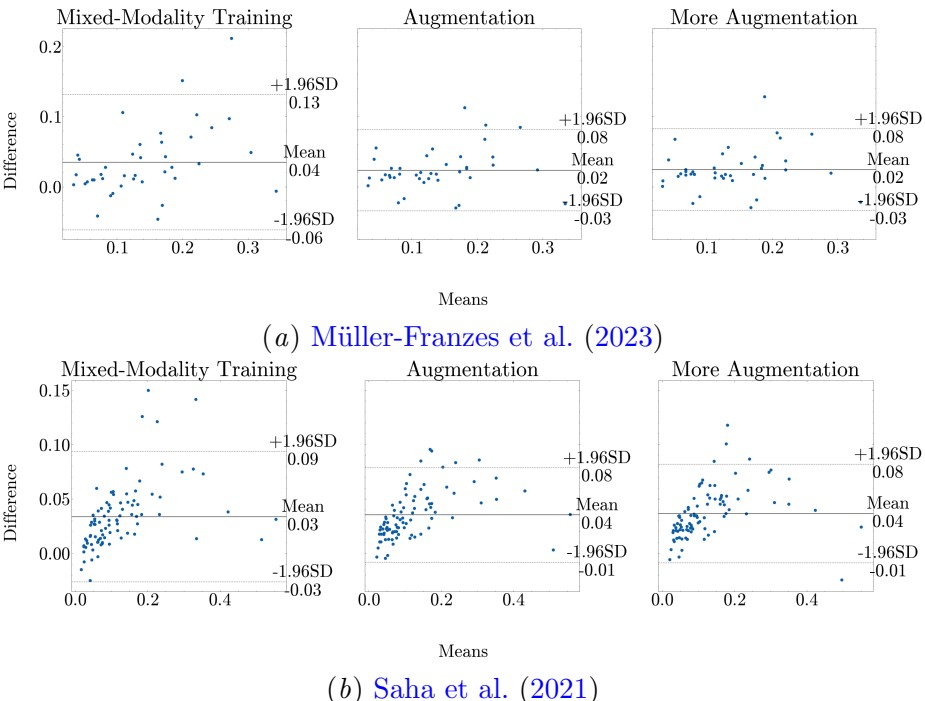

($a$) Müller-Franzes et al. (2023)

($b$) Saha et al. (2021)

Figure 3: Bland-Altman plots visualizing the breast density estimation agreement between predicted and ground truth. The results are in percentage. Means indicate the average between predicted, and ground truth density. Difference is calculated by subtracting predicted from ground truth density.

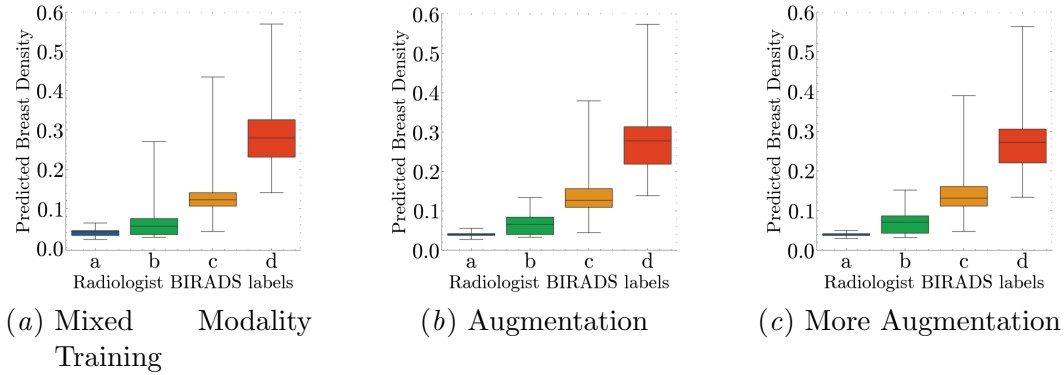

($a$) Mixed Modality Training

($b$) Augmentation

($c$) More Augmentation

Figure 4: Density predicted by network to average BI-RADS assessments by radiologists. Whiskers indicate the 0th, and 100th percentile.

networks trained on style randomization yields more reliable density estimations for all the density labels. This suggests that predicting BIRADS density labels from numerical output is less prone to false positives.

## 4. Discussion

In this study, we explored the effect of style randomization for breast density estimation and proposed the implementation using Dixon images. Experimental results indicate that the network trained with style randomization reduces style bias and forces robustness.

Firstly, style randomization helps with the out-of-domain distribution performance. This finding is inline with Jackson et al. (2019), and Xu et al. (2021). Compared to mixed modality training, we do not observe performance degradation in the FGT Dice score, resolving the style bias observed with mixed modality training. This finding suggests that style randomization may improve the generalization capabilities of the network. Furthermore, correctly segmenting the patient with breast implant may suggest that it forces networks to recognize breast and FGT shape better. This finding is reinforced by significantly higher dice scores with augmentation in independent test sets, and better breast density estimations.

Style randomization improves the breast density estimates both in BIRADS, and numerical estimation tasks. We find that style randomization significantly improves the correlation between predicted and ground truth density. In Figure 3 and 4 we observe that augmentation alleviates the positive outliers. Additionally, the strong agreement between radiologists' assessments and the network predictions reinforces the reliability of the proposed technique. We observe that all networks overshoot the breast density estimation in denser breasts regardless of style randomization. However, this trend is less noticeable with style randomization.

Moreover, we do not observe a statistically significant difference between augmentation strategies in independent test sets. However, we see that using more augmentation further improves the segmentation, and density estimation results in the held-out test set. This finding may emphasize that having more augmentation further reduces the style bias, and enforces more robustness.

There are several limitations of the study. Firstly, we only trained two different augmentation strategies with arbitrarily chosen $\alpha$ values. Comparing more augmentation strategies would better quantify the effect of different style randomization levels. Secondly, training dataset relies on labeled information generated by a pre-trained network. This network over-segments the FGT areas, resulting in lower resolution of FGT segmentation. Networks trained on biased data not only fail to alleviate but can also exacerbate these biases. Consequently, irrespective of the chosen training schema, our model tends to exhibit an over-segmentation tendency in FGT, and may not generate high-resolution FGT segmentations. This may explain the trend we observed with breast density estimations in denser breasts. Denser breasts have more FGT tissue, and thus over segmenting it yields erroneous density estimates.

## 5. Conclusion

Our paper introduces novel style randomization using Dixon's images, demonstrating its efficacy in enhancing network robustness. The proposed technique further reduces style bias, offering a valuable approach for leveraging Dixon images in breast density estimation. Further research may explore style randomization as random data augmentation and training diverse network families to quantify its broader impact.

## Acknowledgments

No acknowledgements.

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
