# OpenReview forum: "Style Randomization Improves the Robustness of Breast Density Estimation in MR Images"
_MIDL.io/2024/Conference — MIDL 2024 Poster_

### Official Review · Reviewer_8Rb8 · 2024-02-28

**Confidence:** 5
**Preliminary Rating:** 5
**Recommendation:** Poster
**Final Rating:** 4

**Summary:**

The work is concerned with breast and fibroglandular tissue (FGT) segmentation in breast MRI for breast density estimation purposes. The novel idea suggested is to interpolate between fat suppressed (FS) and regular T1-weighted images (WOFS) as an augmentation method. Both contrasts can be reconstructed from the same Dixon MR protocol, so they are automatically spatially aligned. The evaluation is done on FS images from an external, public dataset (/ two related ones).

**Strengths:**

The paper is well-written, well-motivated, and the method just makes sense. The evaluation is plausible and reliable and the external test set(s) suitable. Using the nn-UNet framework makes the work more reliable and reproducible.

**Weaknesses:**

The idea is relatively simple and I can imagine that other reviewers would claim lack of novelty. However, given the otherwise solid paper, I think it's worth being published.

The "ground truth" (I personally dislike that term in medical imaging and prefer others such as "reference") is from "a pre-trained network" the provenance of which is unclear. I think this is the strongest weakness of the paper, since the quality of the "ground truth" used for training is

The training set is not very large (n = 51 patients / Dixon acquisitions), so a larger training set could alleviate the need for this kind of augmentation.  However, it is very hard to hand-annotate FGT.

The augmentation is not done on the fly with random mixing factor α (which would have been easily possible also with the nnU-Net / batch generators framework), but only a few fixed uniform steps between 0 and 1.  This also leads to the strange separation between "augmented" (three interpolated steps) and "more augmented" (eight steps), which would otherwise not have been necessary.

**Detailed Comments:**

I think the phrase "style randomization" hints at more and different things than what's actually happening. Maybe "contrast interpolation", "contrast augmentation", "Dixon augmentation" or so would be more fitting (descriptive and not misleading).  In particular, as I mentioned under "weaknesses" above, no randomization is used for training currently.

Also, I think the statement "Domain shift does not affect the performance of the networks trained with augmentations." is too strong and is not justified by the limited experiments.

I think the \cite macro allows for more than one argument, so that  (Dalmı ̧s et al., 2017) (Huo et al., 2021) (Zhang et al., 2019) (Baker et al., 2018) would have just one pair of parentheses.

Figure 2 could be improved.  Also, I suggest including vector graphics, in particular for Figures 2 and 3.  (But also the labels in Figure 1 would benefit from that.)

In 2.1 there is a broken sentence starting with "Where".

In 2.3 there is an "and" too much before "is publicly available".  And "between the periods" should be rephrased.

In the caption of Figure 1, "WOFS" and "FS" seem to be swapped.

In 3, I have problems parsing the sentence "mixed modality training is robust to the WOFS and FS".

3.1 ends with ", ."

3.2 has "box plot of pooled radiologist labels to the density prediction" which I also suggest to rephrase.

A space is missing between "network.This network" in 4.

I suggest changing "over segmentation" into "over estimation", since the former has a different, kind of opposite meaning as well.  (Pet peeve of mine, of course I see that it's commonly used like this in the MIC community though, so you can see this as an opinionated comment independent of the review.)

**Justification Of Final Rating:**

I see that I have been by far the most positive reviewer, but I still like the paper, because it gives a very clear message to the community: If you work with Dixon images, this specific simple data augmentation method is definitely worth implementing. The contribution is small in the sense that it is not very surprising and only affects a niche, but if you happen to be in that niche, it appears to be a solid and reliable (small) contribution, and I like such works.

I am not sure how much I agree with the other reviewer about the mixup comparison – on the one hand, the question "what happens if the mixup is performed inter-modality but also inter-patient or intra-modality and inter-patient?" seems to neglect that fact that the "inter-modality" mixup is the only one that interpolates between two contrasts that by definition *come from the same position* (without any registration). I am also not a particular fan of the concept of mixup augmentation, given that the inputs usually do *not* share the same voxel grid and produce very unreasonably looking inputs. On the other hand, I must agree that the work is close enough to mixup, and the technique is relatively common (well-known and used), so it could have been a good candidate for an ablation study.

In the light of the critical discussion, I slightly change the rating to "weak accept", but my overall impression has not changed much and I still think this is a nice contribution worth being discussed at MIDL.

Finally, I thank the authors for the well-formatted replies to our reviews, which were convenient to read.

**Justification Of The Preliminary Rating:**

Even though the contribution is not large, the paper is well-written and the the method makes sense in the target application, see also "strengths" above. There are weaknesses (again, see above), but they do not affect the main contribution of the paper.

**Questions To Address In The Rebuttal:**

What is the provenance of the model used for FGT "ground truth"?

What do you mean with "[DUKE masks] we could not read" and "were not in the 3D space", respectively?

Why do you use majority voting for the density classes and not some kind of numerical averaging?  Yes, the labels are more ordinal than numerical, but you're already measuring correlation against numerical model outputs.

What do you mean with "we randomly chose two fat fractions and augment *one* image", so why only *one*?

---

> ### Author Response · Authors · 2024-03-16
> **Rebuttal**
>
> We thank the reviewer for their critical assessment of our work. In the following we address their major concerns point by point.
>
>
> __Reviewer Point 1__ — What is the provenance of the model used for FGT ”ground truth”?
>
> The model used for the ground truth is a pretrained model on a small subset of our data with manual
> segmentations of the breast and FGT. This model was used to generate our training set. We manually
> inspected the network-generated segmentations and checked for inaccuracies, excluding segmentations
> with obvious mistakes.
> ***
> __Reviewer Point 2__ — What do you mean with ”[DUKE masks] we could not read” and
> ”were not in the 3D space”, respectively?
>
> There were segmentation images that had more than three dimensions. Because there was
> no metadata available for us to parse these we decided to discard them. Moreover, we utilized SITK
> to read the segmentations, and some could not be read. We decided to discard them due to lack of
> metadata as well.
> To address this weakness, we will make sure to make this explicit in the paper.
> ***
> __Reviewer Point 3__ — Why do you use majority voting for the density classes and not some
> kind of numerical averaging? Yes, the labels are more ordinal than numerical, but you’re already
> measuring correlation against numerical model outputs.
>
> We indeed used majority voting because the density classes are ordinal. It turns out that there
> would be no difference between averaging or majority voting as in all cases two out of three radiologists
> agreed on the density grade, with the third radiologist estimate differing at most one grade from the
> other.
> We are planning to add further explanation in the methodology section regarding the average pooling,
> and used BIRADS labels.
> ***
> __Reviewer Point 4__ — What do you mean with ”we randomly chose two fat fractions and
> augment one image”, so why only one?
> Mathematically.
>     \begin{align*}
>         &\alpha_{1} \sim Uniform(0,1)\\
>         &\alpha_{2} \sim Uniform(0,1)\\
>         &I_{1} = W_{1} + \alpha_{1} * F_{1}\\
>         &I_{2} = W_{1} + \alpha_{2} * F_{1}
>     \end{align*}
>
> We use these to create random augmentation of the test data. Our Test Data contains 18 images [9WOFS, 9FS]. To reach to the same number of cases in our style randomized test data, we augment one image using two fat fractions. It is possible to choose more alpha values, however we decided to go with two for the aforementioned reason.

---

### Official Review · Reviewer_JzNm · 2024-02-28

**Confidence:** 4
**Preliminary Rating:** 3
**Recommendation:** Poster

**Summary:**

To alleviate the style bias inherent to segmentation models trained on multi-modality MRI, the authors propose to use style randomization as a data augmentation strategy. They conduct experiments against a baseline and a second augmentation method to show that their proposed approach outperforms both in terms of generalisation/robustness and consequently, accuracy.

**Strengths:**

* Trained in-house, validated on DUKE dataset -- those are probably reasonably different, which helps to underscore the claim the paper makes.
* Overall fair quality
* Despite partial shortcomings in English, overall very easy to follow.
* largely good and self-critical discussion of the achievements

**Weaknesses:**

* Style randomization is a catchy word for "linear interpolation between Fat and Water contrast...
* Ground truth for the parenchyma masks was another algorithm, introducing a bias.
* Ground truth for BI-RADS labels were likely physician-generated, and might be highly variable.
* Claiming that augmentation leads to robustness, I would have expected a validation that shows this: It should show the improvement in relation to the baseline, ie. it should improve the worst baseline cases the most. I think this is an opportunity missed.
* In your comment to figure 4, what do you mean when you say the network "overestimates" the density? Isn't it only important to be able to map the numerical estimate to the proper score (define thresholds)? In fact, looking at the whiskers of group D in the augmented scenarios, they reach into the box area, which they don't for the "mixed modality" scenario -- setting a threshold causes less group confusion! The same holds for the situation between groups B and C, which is the dividing line between "not so dense" and "fairly dense" in a binary decision scenario. This would actually mean that the mapping from the numerical density to the score suffers from the augmentation.

**Detailed Comments:**

* Page 2, top paragraph: the paper from Baker 2018 is presented to use results from the paper of Zhang 2019, which is impossible if the years can be trusted.
* Figure 1 caption: alpha 0 corresponds to the water image (FS), alpha 1 to the fat image (WOFS). That's mixed up in the caption.
* page 4 bottom: augmentation helps... up to 0.4 -- I don't understand this. improves by 0.4 or leads to a DICE of 0.4?
* Does Figure 2 also show that for some cases, the augmentation deteriorates the DICE? Please explain.
* Table 2 should be called a Figure.

**Justification Of The Preliminary Rating:**

The basic idea isn't excessively novel, the datasets are not really large, and the improvements are large only in a few cases (well, of course a consequence of the small datasets).
It's a simple to implement strategy for a quite particular purpose and modality, so nothing wrong about presenting it, just perhaps not quite in an oral.

**Questions To Address In The Rebuttal:**

All above ;-)

**Special Issue:**

No

---

> ### Author Response · Authors · 2024-03-16
> **Rebuttal**
>
> We thank the reviewer for their critical assessment of our work. In the following we address their major concerns point by point.
>
> __Reviewer Point 1__ — Style randomization is a catchy word for ”linear interpolation between
> Fat and Water contrast...
>
> Our paper introduces a new way to augment training data using the linear interpolation
> between fat and water images. We extend on the idea of using Dixon images for multi modality
> training.  We argue that this falls into the category of style randomization as the ”style” of the image change
> while keeping the global structure of the image. Figure 1 shows this phenomenon. Moreover referring
> to our method as ”style randomization” places the paper in a more broad scientific context.
> However, we do agree that in this case style randomization boils down to relatively simple linear
> interpolation.
> ***
> __Reviewer Point  2__ — Ground truth for the parenchyma masks was another algorithm,
> introducing a bias.
>
> We are aware of this limitation, and discuss this in the last paragraph of discussion section on
> page 8.
>
> > Secondly, training dataset relies on labeled information generated by a pre-trained network.This network over-segments the FGT areas, resulting in lower resolution of FGT segmentation. Networks trained on biased data not only fail to alleviate but can also exacerbate these biases ...
> ***
> __Reviewer Point 3__ — Ground truth for BI-RADS labels were likely physician-generated,
> and might be highly variable.
>
> Yes, there are variances to the BIRADS density grades, however in this case two out of three
> radiologists always agreed on a grade. And, the odd one out was always one label level off. We also
> used average pooling to reduce the variance as much as possible. We will add further explanation in
> the methodology section to address this weakness.
> ***
> __Reviewer Point 4__ — Claiming that augmentation leads to robustness, I would have expected
> a validation that shows this: It should show the improvement in relation to the baseline, ie. it should
> improve the worst baseline cases the most. I think this is an opportunity missed.
>
> Tables 1 and 2 indicate this phenomenon where there is improvement over baseline. Furthermore, Table 2 indicates that the cases that are predicted poorly by mixed modality training are improved by using style randomization. One of the most interesting case comes from MR Image with Breast Implant. Without using style randomization, we see a bad segmentation of breast and FGT tissue, however
> with proposed augmentation this phenomenon is fixed and the segmentations are more reliable.
> Moreover, Figure 2 supports our claims as well. In the sub figure (a) it is shown that compared to
> mixed modality training, augmentation improves a case by 0.40 dice score . Moreover, subfigure (b)
> shows that it improves a case by 0.25 dice score.
>
> We also discuss why augmentation could lead to robustness in the discussion section.
> Furthermore, correctly segmenting the patient with breast implant may suggest that it forces
> networks to recognize breast and FGT shape better. This finding is reinforced by significantly
> higher dice scores with augmentation in independent test sets, and better breast density
> estimations
>
> To address this weakness, we plan to comment on Table 2 further with respect to ”it should improve
> the worst baseline cases the most”.
> ***
> __Reviewer Point 5__ — In your comment to figure 4, what do you mean when you say the
> network ”overestimates” the density? Isn’t it only important to be able to map the numerical
> estimate to the proper score (define thresholds)? In fact, looking at the whiskers of group D in
> the augmented scenarios, they reach into the box area, which they don’t for the ”mixed modality”
> scenario – setting a threshold causes less group confusion! The same holds for the situation between
> groups B and C, which is the dividing line between ”not so dense” and ”fairly dense” in a binary
> decision scenario. This would actually mean that the mapping from the numerical density to the
> score suffers from the augmentation.
>
> When network predicts density estimation that is an outlier for the BIRADS density distribution, we call this ”over-estimation”. We show that over-estimation is less prominent when we use
> style randomization. Our method pushes edge cases towards the median of the BIRADS label density
> distribution. Furthermore, figure 4 shows that augmentation leads to more reliable density predictions.
>
> Augmentation actually causes less group confusion, because the predictions that are over-estimated
> were placed in correct vicinity, however this also makes the tails longer. By looking at the outliers that
> are alleviated, the reason why whiskers reach to group ”D” is due to ”lowering” the predictions of the
> outlier cases to fit into the tails.
> To take the stated weakness into account, we plan to further explain figure 4 in the results section.

---

### Official Review · Reviewer_tLkN · 2024-03-05

**Confidence:** 3
**Preliminary Rating:** 2
**Final Rating:** 2

**Summary:**

This paper proposes a data augmentation strategy/style randomization technique mixing different MRI modalities which results in more accurate breast and fibro-glandular tissue (FGT) segmentation on Breast MRI. This enables more precise breast density estimation. The authors performed training on an in-house dataset and evaluated the performances of the semantic segmentation model on diverse datasets (in-house and independent test sets).

**Strengths:**

- Designing better and more robust approaches for breast density estimation is important and useful for the medical community.
- The paper is quite clear and well-written.
- Even though, the authors have performed the training on an in-house dataset and hence results are not reproducible, they have evaluated the proposed method on independent test sets and obtained robust performances.

**Weaknesses:**

* The originality/novelty of the work is somewhat limited. Even though the mixing strategy is performed differently than usual (intra-patient and inter-modality). The paper seems to be a very slight adaptation of the mixup data augmentation for a breast MRI segmentation task. Using mixup augmentation in the context of MRI segmentation tasks is already common like in [1].
* The paper lacks ablation studies and other comparisons with other augmentation strategies:
     * For instance, what happens if the mixup is performed inter-modality but also inter-patient or intra-modality and inter-patient?
     * How does this method compare to classic and powerful augmentation/style randomization strategies such as Fourier-based augmentation [2]?

References:
* [1] Panfilov, E., Tiulpin, A., Klein, S., Nieminen, M. T., & Saarakkala, S. (2019). Improving robustness of deep learning based knee mri segmentation: Mixup and adversarial domain adaptation. In Proceedings of the IEEE/CVF International Conference on Computer Vision Workshops (pp. 0-0).
* [2] Xu, Q., Zhang, R., Zhang, Y., Wang, Y., & Tian, Q. (2021). A fourier-based framework for domain generalization. In Proceedings of the IEEE/CVF Conference on Computer Vision and Pattern Recognition (pp. 14383-14392).

**Detailed Comments:**

No specific comments.

**Justification Of Final Rating:**

I thank the authors for their answers as it has clarified some points of the paper. I still think that the paper lacks experiments using different mixup settings (inter patient - Intra modality, Inter patient - Inter modality). The proposed method is almost a direct application of mixup on opposing modalities for a very specific segmentation problem, I think it is important that the paper explores the different possible mixup settings as It would prove that the proposed mixing strategy is optimal for this problem. Since, no experiments have been carried out to compare the different possible mixing strategies, I will keep the grade as it was.

**Justification Of The Preliminary Rating:**

Given the limited novelty of this work, the lack of ablation studies regarding the mixup augmentation, and missing comparisons with other data augmentation strategies, I give a weak reject grade. However, I am willing to increase the grade if the authors address my concerns.

**Questions To Address In The Rebuttal:**

Please address concerns in the weaknesses section.

**Special Issue:**

No

---

> ### Author Response · Authors · 2024-03-16
> **Rebuttal**
>
> We thank the reviewer for their critical assessment of our work.
> In the following we address their major concerns point by point.
>
> __Reviewer Point 1__ — The originality/novelty of the work is somewhat limited. Even though
> the mixing strategy is performed differently than usual (intra-patient and inter-modality). The
> paper seems to be a very slight adaptation of the mixup data augmentation for a breast MRI
> segmentation task. Using mixup augmentation in the context of MRI segmentation tasks is already
> common like in [1].
>
> Thank you for providing further resources for the paper. We’d like to acknowledge that our
> method is a mix-up between Fat and Water MR Images. However, the proposed mix-up has a special
> property of linearly mixing two modalities when used in intra-patient and inter-modality setting.
> The FS and WOFS images can be viewed as two opposing sides of fat content distribution. FS has
> no fat, and WOFS has all the fat. Using the proposed augmentation, we then can generate images that
> look more like WOFS, or FS. Or, images that do not look like either. In page 3 figure 1, we demonstrate
> this effect.
> By using mix-up this way, we force the network to focus more on the FGT, and breast rather than
> the fat content in the image. Moreover, the methodology in paper [1] was not applied on Dixon Images,
> therefore it does not have the special property of generating from the distribution of two opposing
> modalities.
> Our contribution relies on realising that the use mix-up with Dixon Images could generate possible
> MR acquisitions from two opposing modalities, and further. This way we explore the realm of MR
> Images with any fat content, and we can go beyond WOFS, and FS domain. Training the segmentation
> network with the generated data results in robustness properties. We think this is also refereed as
> ”Domain Generalisation” in the literature.
>
> ***
> __Reviewer Point 2__— For instance, what happens if the mixup is performed inter-modality
> but also inter-patient or intra-modality and inter-patient?
> Reply: As explained above, we achieve linearly mixing two domains when we perform Dixon augmentation on inter-modality and intra-patient setting. Therefore, we think that it is not possible to use this
> method for other possible settings, and have performance gains.
>
> As explained above, we achieve linearly mixture of two modalities when we perform Dixon augmen-
> tation on inter-modality and intra-patient setting. Therefore, we think that it is not possible to use this
> method for other possible settings, and have performance gains
>
> ***
> __Reviewer Point 3__ — How does this method compare to classic and powerful augmentation/style randomization strategies such as Fourier-based augmentation [2]?
>
> We thank the reviewer for providing an interesting paper on the augmentation literature. We
> believe that our proposed method can be viewed as a special case of Fourier augmentation method. In
> the paper [2], authors propose to mix amplitude signals of two images linearly, while keeping the phase
> signal unchanged. They argue that the phase signal corresponds to the global object structure, thus using such augmentation teaches network to focus more on the global shape rather then local texture.
> In our method, we augment water images with the fat content. Due to the reconstruction of water
> and fat images, they will have the same global structure, but different local textures. Therefore, in our
> method, we are also keeping the global structure intact while mixing the local textures. However, we
> deviate from the Fourier transformation by augmenting the local texture where it is necessary. The fat
> and water signals are found in the breast’s and the chest. And, that is the most interesting part when
> we perform the segmentation task. Moreover, our method is capable of augmenting only local textures
> in this area without any need of prior segmentations to mask the region of interest. We will include
> this paper in the related research, and discuss the findings with respect to aforementioned paper’s view
> point as well as comparing our methodologies.
> ***
> __Reviewer Point 4__ — The paper lacks ablation studies and other comparisons with other
> augmentation strategies
>
> Our paper’s main focus is on developing a method that can be used to force the network
> to be more robust on the shape rather then the texture. We carried out an ablation study for the
> level of augmentation to understand the change in performance with respect to the augmentation level.
> Moreover, we compared it with solely training on mixed modality setting as well.

---

> > ### Comment · Reviewer_tLkN · 2024-03-21
> >
> > I thank the authors for their answers as it has clarified some points of the paper. I still think that the paper lacks experiments using different mixup settings (inter patient - Intra modality, Inter patient - Inter modality). Since the proposed method is almost a direct application of mixup on opposing modalities for a very specific segmentation problem, I think it is important that the paper explores the different possible mixup settings. It would prove that the proposed mixup strategy is optimal for this problem.

---

### Author Response · Authors · 2024-03-16
**We thank the reviewers for assesing our paper**

In the rebuttal we address their major concerns point by point.
Furthermore, we address the detailed comments regarding the language, citations, and figures in the revised version of our paper.

---

### Comment · Area_Chair_kegB · 2024-03-19
**Paper is open for dsicussions**

Dear Reviewers The authors have submitted their rebuttal addressing the raised questions. The paper remains open for further discussion and engagement.

---

### Meta-Review · Area_Chair_kegB · 2024-04-03

**Recommendation:** Accept (Poster)
**Confidence:** 5

**Metareview:**

Two of the three reviewers support accepting this work as a poster and believe it will generate meaningful discussions. I believe the rebuttal has addressed most of the important comments. The authors are requested to revise the camera-ready version in accordance with the feedback provided in the rebuttal.

---

### Decision · Program_Chairs · 2024-04-06

Accept (Poster)